# Opposing Effects of Interleukin-36γ and Interleukin-38 on Trained Immunity

**DOI:** 10.3390/ijms24032311

**Published:** 2023-01-24

**Authors:** Lisa U. Teufel, Mihai G. Netea, Frank L. van de Veerdonk, Charles A. Dinarello, Leo A. B. Joosten, Rob J. W. Arts

**Affiliations:** 1Department of Internal Medicine, Radboud Institute of Molecular Life Sciences (RIMLS) and Radboudumc Center for Infectious Diseases (RCI), Radboud University Medical Center, 6525 GA Nijmegen, The Netherlands; 2Department of Immunology and Metabolism, Life and Medical Sciences Institute, University of Bonn, 53115 Bonn, Germany; 3Department of Medicine, University of Colorado, Aurora, CO 80045, USA; 4Department of Medical Genetics, Iuliu Hatieganu University of Medicine and Pharmacy, 400001 Cluj-Napoca, Romania

**Keywords:** IL-36, IL-38, innate immune memory, epigenetics, immunometabolism

## Abstract

Trained immunity is the process of long-term functional reprogramming (a *de facto* innate immune memory) of innate immune cells such as monocytes and macrophages after an exposure to pathogens, vaccines, or their ligands. The induction of trained immunity is mediated through epigenetic and metabolic mechanisms. Apart from exogenous stimuli, trained immunity can be induced by endogenous compounds such as oxidized LDL, urate, fumarate, but also cytokines including IL-1α and IL-1β. Here, we show that also recombinant IL-36γ, a pro-inflammatory cytokine of the IL-1-family, is able to induce trained immunity in primary human monocytes, demonstrated by higher cytokine responses and an increase in cellular metabolic pathways both regulated by epigenetic histone modifications. These effects could be inhibited by the IL-36 receptor antagonist as well as by IL-38, an anti-inflammatory cytokine of the IL-1 family which shares its main receptor with IL-36 (IL-1R6). Further, we demonstrated that trained immunity induced by IL-36γ is mediated by NF-κB and mTOR signaling. The inhibitory effect of IL-38 on IL-36γ-induced trained immunity was confirmed in experiments using bone marrow of IL-38KO and WT mice. These results indicate that exposure to IL-36γ results in long-term pro-inflammatory changes in monocytes which can be inhibited by IL-38. Recombinant IL-38 could therefore potentially be used as a therapeutic intervention for diseases characterized by exacerbated trained immunity.

## 1. Introduction

Trained immunity is the establishment of a non-specific memory of innate immune cells. When cells of the innate immune system, typically monocytes, are stimulated with a certain ligand or pathogen, epigenetic and metabolic changes are induced which results in an enhanced response upon restimulation with heterologous stimuli [1,2]. The induction of trained immunity was first described by stimulation of monocytes with *Candida albicans* or its cell wall component β-glucan, and by the BCG vaccine or its peptidoglycan/muramyl dipeptide component [3,4]. In the meantime, several other pathogens and ligands have been shown to induce trained immunity. Interestingly, apart from pathogens and their ligands, cytokines have also been reported to induce innate immune memory. It was first shown that interleukin-1β (IL-1β) possesses this potential [5], and later it was shown that IL-37 and IL-38, anti-inflammatory cytokines of the IL-1-family, are able to inhibit the induction of trained immunity by β-glucan in mice [6,7].

The pro-inflammatory cytokine IL-36 exists in three agonistic isoforms, α, β, and γ, that are functionally antagonized by the anti-inflammatory receptor antagonist IL-36Ra. IL-36 binds to the IL-36 receptor (IL-1R6) and its co-receptor IL-1 receptor accessory protein (IL-1RAcP). Upon recruitment of the co-receptor, an intracellular complex of two TIR domains is assembled which results in the downstream activation of MAPK and NF-κB signaling [8]. IL-36 also belongs to the larger IL-1 superfamily and was shown to play a role in several inflammatory conditions [9]. Importantly, IL-38 also binds to the IL-1R6 to induce its anti-inflammatory effects [10]. Given the essential role of other IL-1-family members in the induction of trained immunity, we aimed to assess the role of IL-36 in trained immunity.

## 2. Results

Human primary monocytes were incubated for 24 h with BCG or β-glucan with or without 1 h pre-incubation with IL-36 receptor antagonist. After washing away the stimuli, cells were left to rest in medium supplemented with 10% human serum. After five days, cells were restimulated with LPS for 24 h, and cytokine production was assessed in the supernatant. TNF and IL-6 production was upregulated in cells previously exposed to either BCG or β-glucan (Figure 1A), a standard model of trained immunity [11]. Addition of IL-36RA to this experiment resulted in inhibition of the induction of trained immunity (Figure 1A). When IL-38 was used instead of IL-36RA, a comparable pattern was observed (Figure 1B). These results suggest that IL-36 may be involved in the induction of trained immunity. Therefore, gene expression of IL-36A, B, and G was determined 4 h after stimulation with BCG and β-glucan, all of which were indeed induced by both stimuli (Figure 1C).

Next, to validate the role of IL-36 in trained immunity, we used one of the isoforms (IL-36γ) to induce trained immunity. When human monocytes were exposed for 24 h to IL-36γ and restimulated with LPS on day six, we observed an increased production of TNF and IL-6 compared to non-treated cells, showing that IL-36γ is able to induce trained immunity (Figure 2A). Addition of IL-36RA during the first 24 h resulted in the complete inhibition of this effect (Figure 2A). Moreover, when IL-38 was used instead of IL-36RA, the upregulation of cytokine production seen during the induction of trained immunity was counteracted in a similar manner (Figure 2B).

To further substantiate the role of IL-36 and IL-38 in trained immunity, we performed in vitro experiments on bone marrow of WT, IL-38KO, and IL1R9KO (the co-receptor for IL-38 signaling [12]) mice. Firstly, bone marrow of WT mice was stimulated with β-glucan or IL-36γ in the presence or absence of IL-38 for 48 h. After four days of rest, cells were restimulated with LPS for 24 h. Unstimulated cells did not produce cytokines, while an increase in TNF production was observed in both cells exposed to β-glucan or IL-36γ (Figure 2C). This effect was inhibited by adding IL-38 (Figure 2C) comparable to training of human monocytes. Next, fresh bone marrow cells of WT and IL-38KO mice was trained with IL-36γ, resulting in an increased induction of TNF production in the IL-38KO mice compared to the WT mice (Figure 2D). Finally, trained immunity was induced by IL-36γ in fresh bone marrow cells of IL-38KO and IL1R9KO mice (Figure 2E). Induction of trained immunity by IL-36γ could be inhibited by supplementation of IL-38 in the IL-38KO mice, whereas addition of IL-38 was ineffective in bone marrow of the IL1R9KO animals (Figure 2E).

To further examine the properties of IL-36γ in trained immunity, we assessed markers of open chromatin which have been associated with trained immunity [1]. Firstly, it was shown that transcription of *TNFA* and *IL6* was induced in IL-36γ-trained cells (Figure 2F), and next, that trimethylation of lysine 4 at histone 3 (H3K4me3) was increased at the promoters of both genes (Figure 2G).

Another important mechanism essential for the induction of trained immunity, which is regulated by epigenetic changes, is cellular metabolism [1]. Therefore, transcription and H3K4me3 at the promoter regions of two rate limiting enzymes in the glycolysis pathway, hexokinase 2 (*HK2*) and phosphofructokinase P (*PFKP*), were assessed. Expression and H3K4me3 of both genes were upregulated by IL-36γ (Figure 3A,B), which is a known phenotype of trained immunity [13,14]. In line with previous results, exposure to IL-38 inhibited this effect. To determine whether this corresponds to increased glycolytic activity of these cells, lactate production after restimulation with LPS on day six was assessed. IL-36γ stimulation during the first 24 h of the experiment induced increased lactate production six days later, which was inhibited by IL-36RA and IL-38 (Figure 3C). To further assess the metabolic activity of IL-36γ-trained monocytes, we performed a Seahorse analysis. It has been reported that β-glucan-trained monocytes show a typical Warburg effect after seven days, characterized by an increased glycolysis and decreased oxidative phosphorylation, whereas other training stimuli such as BCG induce an increase in both glycolysis and oxidative phosphorylation [14,15]. IL-36γ-trained monocytes also show an induction of both glycolysis and oxidative phosphorylation after seven days (Figure 3D,E). Interestingly, these metabolic effects could also be inhibited by the addition of IL-38 during the first 24 h.

To further assess the mechanisms through which IL-36γ induces trained immunity, we investigated downstream signaling pathways associated with the induction of innate immune memory. It has been shown previously that mTOR is one of the central regulatory molecules in the induction of trained immunity [7,15]. IL-36γ is known to signal via MyD88, MAP kinases, and NF-κB [8], and IL-38 is able to inhibit MAP kinases and also mTOR signaling [7,16]. Here, we now also show that IL-36γ induces mTOR activation which can be inhibited by the addition of IL-38 (Figure 4A). In order to expand this view on essential signaling molecules in the induction of trained immunity, we next focused on NF-κB. Firstly, human monocytes were pre-incubated for 1 h with IL-38, before IL-36γ was added to the medium. After 10 min, cells were lysed, and phosphorylation of NF-κB was determined by Western blot. IL-36γ increased p65 phosphorylation of the NF-κB complex, and pre-incubation with IL-38 abrogated this effect (Figure 4B). To determine whether NF-κB activation is indeed essential in the induction of trained immunity, human monocytes were incubated for 24 h with β-glucan, BCG, and IL-36γ in combination with an inhibitor of NK-κB activation. After five days of rest in 10% human serum, cells were restimulated with LPS, and cytokine production was assessed in supernatants. Both TNF and IL-6 production was reduced by the inhibition of NK-κB activation (Figure 4C), validating its importance for the induction of trained immunity.

## 3. Discussion

Trained immunity entails the establishment of a non-specific memory of an insult on cells of the innate immune system, which can be induced by various pathogens, vaccines, and their ligands. Interestingly, apart from microbial ligands, certain cytokines are able to induce trained immunity. We have now shown that IL-36γ also induces a trained immunity phenotype in human myeloid cells, a process that can be inhibited by its receptor antagonist as well as the anti-inflammatory cytokine IL-38. These results were confirmed in murine bone marrow cells from WT, IL-38KO, and IL-1R9KO animals.

IL-36γ is able to induce all typical properties of trained immunity. Cytokine production is increased after restimulation with heterologous compounds in cells trained for six days with IL-36γ, and this process is regulated by epigenetic modifications. Further, the cellular metabolism is changed towards heightened glycolysis and oxidative phosphorylation, and likewise modulated by epigenetic changes. Finally, we show a role of mTOR and NF-κB in IL-36γ-induced trained immunity.

By assessing the downstream pathways engaged by both established training stimuli and the IL-36 subfamily, we observe an overlap between the kinases required for the induction of trained immunity and the signaling pathway of IL-36. IL-36γ binds to the IL-36 receptor which results in two distinct intracellular cascades. Through MyD88 and IRAK signaling, this results in either the activation of MAP kinases or NF-κB signaling, which in turn induces gene expression [17]. Moreover, IL-36γ has been shown to be an inducer of autophagy [18], which has been reported to be an essential regulator, especially of BCG-induced trained immunity [19]. Here, we demonstrate a direct effect on the induction of training by IL-36γ through the activation of NF-κB in human monocytes. Moreover, mTOR is a central cellular regulator of trained immunity, among others in the induction of cellular metabolism [15]. Previously, IL-36γ has already been shown to cause mTOR phosphorylation in other cell types [20]. We now also show that IL-36γ has this effect in human monocytes.

IL-36 is mainly produced in epithelial tissues such as keratinocytes, but also in monocytes, and the dysregulated production of IL-36 has been associated with several diseases. Increased IL-36 concentrations are, e.g., linked to psoriasis, but also auto-immune diseases such as systemic lupus erythematosus (SLE), inflammatory bowel disease, and Sjögren’s disease [21]. Furthermore, IL-36γ has been shown to aggravate foam cell formation and atherosclerosis [22], and a rare deficiency of the IL-36 receptor antagonist (DITRA) results in a disease with generalized pustular psoriasis with systemic inflammation [23]. We now show that the stimulation of monocytes with IL-36γ causes epigenetic changes which result in a more pro-inflammatory phenotype of these cells, based on a more active cellular metabolism and an increased cytokine response to a secondary stimulus. Therefore, it seems likely that IL-36 plays a central role in several inflammatory diseases (infectious, auto-immune, or auto-inflammatory) and that inhibition of IL-36, either by its receptor antagonist or by IL-38, is of potential interest to modulate inflammation and disease activity.

## 4. Materials and Methods

### 4.1. Monocyte Isolation

Monocytes were isolated from buffy coats, obtained from four to six healthy donors, depending on the experiment from Sanquin Blood Bank, Nijmegen, the Netherlands. Firstly, peripheral blood mononuclear cells (PBMCs) were isolated by differential density centrifugation over Ficoll-Paque (GE healthcare, Buckinghamshire, UK) of blood diluted 1:5 in sterile, pyrogen-free phosphate-buffered saline (PBS). Cells were washed three times with PBS and quantified using a haematology analyser (XN-45 haematology analyser; Sysmex Corporation, Kobe, Japan). Percoll isolation of monocytes was performed as described previously [11,24]. Briefly, PBMCs were put on a hyper-osmotic Percoll solution (48.5% Percoll [Sigma-Aldrich, St. Louis, MO, USA], 41.5% sterile H_2_O, 0.16 M filter-sterilized NaCl) and centrifuged for 15 min at 580× *g* (4 °C) before collecting the interphase layer containing the monocytes. Cells were washed once with cold PBS, suspended in Dutch modified RPMI 1640 culture medium (Invitrogen, Carlsbad, CA, USA) supplemented with 5 μg/mL gentamicin, 2 mM Glutamax (Gibco, Carlsbad, CA, USA) and 1 mM pyruvate (Gibco, Carlsbad, CA, USA), and quantified (Sysmex Corporation).

### 4.2. Induction of Trained Immunity in Human Monocytes In Vitro

Trained immunity was induced in an in vitro model, as described previously [11]. Briefly, monocytes were obtained as described above and plated in flat-bottom 96-well plates, with 10^5^ cells per well, and in some experiments, pre-incubated with IL-38 (3-152, 90 ng/mL; R&D systems, Minneapolis, MN, USA), IL-36RA (400 ng/mL; PeproTech, London, UK), or an InSolution NF-κB activation inhibitor (100 nM; Calbiochem, San Diege, CA, USA) for 1 h at 37 °C, or with supplemented RPMI, as a negative control. Subsequently, cells were trained for 24 h at 37 °C using Bacillus Calmette-Guerin (BCG; 2 μg/mL; SSI, Copenhagen, Denmark), β-1,3-(D)-glucan (2 μg/mL; kindly provided by Professor David Willems, Johnson City, USA), and IL-36γ (50 ng/mL; PeproTECH) using supplemented RPMI as a negative control. After 24 h, cells were washed with warm PBS and incubated in supplemented RPMI with 10% human serum for five days. On day six, cells were restimulated for 24 h with lipopolysaccharide (LPS) (10 ng/mL) from *E. coli* serotype O55:B5 (Sigma-Aldrich, Darmstadt, Germany). Supernatants were stored at −20 °C until further analysis.

### 4.3. Induction of Trained Immunity in Murine Bone Marrow In Vitro

Fresh bone marrow cells were isolated from one femur from eleven months old, male, wild type (WT), IL-38 knock out (KO), and IL1R9KO mice. The experiments were approved by the Institutional Animal Care and Use Committees of the University of Colorado Denver, Aurora, CO (protocol # 00105). Bone marrow was aseptically removed, filtered through 100 μm cell strainer (Fisher scientific, Carlsbad, CA, USA), and collected in RPMI. Cells were counted, and 5 × 10^5^ cells were seeded in flat-bottom 96-well plates in a final volume of 200 μL of DMEM (Invitrogen) supplemented with 10% fetal bovine serum (Thermofisher Scientific, Carlsbad, CA, USA) and penicillin/streptavidin (Gibco, Carlsbad, CA, USA). During the first 48 h, 5 μg/mL β-glucan, 50 ng/mL IL-36γ, or medium as a control was added with or without 90 ng/mL IL-38. After 48 h, cells were washed with 200 μL of warm PBS to wash away the stimuli and non-adherent cells. Cells were left to rest in supplemented DMEM for four days, after which, cells were restimulated with 10 ng/mL LPS for 24 h. Supernatants were stored at −20 °C until further analysis.

### 4.4. ELISA and Lactate Assay

IL-6, TNF (both human and mouse), and IL-36 concentrations were measured in cell culture supernatants by DuoSet ELISA kits (R&D Systems, Minneapolis, MN, USA) according to manufacturer’s instructions. Lactate concentrations were measured in cell culture supernatants on day seven of the training protocol using a lactate fluorometric assay kit (BioVision, Waltham, MA, USA). Culture medium was used as a negative control.

### 4.5. Quantitative RT-PCR

Monocytes were isolated and trained as described above. RNA was isolated after 4 h on the first day or after 4 h on day seven after LPS restimulation. RNA was isolated by an RNeasy column isolation kit (Qiagen GmbH, Hilden, Germany) according to the manufacturer’s instructions. RNA was eluted in Rnase-free water, and cDNA was subsequently synthesized by reverse transcription using iScript (Bio-Rad, Hercules, CA, USA). Diluted cDNA was used for qPCR performed by the StepOne PLUS sequence detection system (Applied Biosystems, Foster City, CA, USA) using SYBR Green Mastermix (Applied Biosystems, Foster City, CA, USA). Gene expression was normalized to the housekeeping gene *β2M* for untrained cells and *HPRT* for cells with a trained phenotype, and analysis was performed following a comparative Ct method. Primer sequences are listed in Table 1.

### 4.6. Metabolic Analysis

Monocytes were isolated as described above. Monocytes (15 × 10^6^ cells/5 mL per dish) were left to adhere on 10 cm polystyrene petri dishes (Corning, Corning, NY, USA) for 1 h at 37 °C. Subsequently, trained immunity was induced as described above. On day six before restimulation with LPS, cells were harvested after detachment by incubation for 30 min with Versene (Gibco, Carlsbad, CA, USA). Cells were counted, and 1 × 10^5^ cells per well were plated in quintuplicates in 96-well Seahorse plates in Seahorse medium (DMEM supplemented with 1 mM pyruvate, 2 mM L-glutamine, and 11 mM D-glucose for OCR and supplemented with 1 mM glutamine for ECAR; pH adjusted to 7.4) and left to adhere for 1 h at 37 °C in a non-CO2-corrected incubator. The assay was performed in a cartridge that has been calibrated overnight in Seahorse calibration medium. Oxygen consumption rate (OCR) and extracellular acidification rate (ECAR) were measured in an XFp Analyzer (Seahorse Bioscience, Agilent, USA)) with final concentrations of 1 μM oligomycin A, 1 μM FCCP, 1.25/2.5 μM rotenone/antimycin A, 11 mM glucose, and 22 mM 2-deoxyglucose.

### 4.7. Epigenetic Analyses (ChIP-qPCR)

Monocytes were isolated, cultured, and harvested as described for the metabolic analysis. Cells were cross-linked in methanol-free 1% formaldehyde (Sigma-Aldrich, Darmstadt, Germany) and 125 μM of glycine. Fixed cell preparations were sonicated using a Diagenode Bioruptor for ten rounds (30 s on and 30 s off) followed by immunoprecipitation. Sonicated chromatin was stored at −80 °C until further analysis. Half of the chromatin of 1 × 10^6^ cells was incubated overnight with an H3K4me3 antibody (1 μg; Diagenode, Seraing, Belgium) to yield H3K4me3-positive chromatin, and the other half was used as total input. After washing the antibody away, DNA from both samples was purified over DNA purification columns (Qiagen GmbH, Hilden, Germany), and qPCR was performed to determine the percentage of H3K4me3 DNA for promoters of the genes in Table 2. *MYO* was used as a positive and *H2B* as a negative control.

### 4.8. Western Blots

Monocytes were added to 1.5-mL Eppendorf tubes. Then, 2 × 10^6^ monocytes were pre-incubated for 1 h with IL-38, before adding IL-36. After 10 min, cells were lysed and stored at −20 °C. The amount of protein was assessed using a BCA assay (Thermofisher Scientific, Carlsbad, CA, USA), and equal amounts of proteins were loaded on pre-casted 4–15% gels (Bio-Rad, Hercules, CA, USA). The separated proteins were transferred to a nitrocellulose membrane (Bio-Rad, Hercules, CA, USA), which was blocked in 5% BSA (Sigma-Aldrich, Darmstadt, Germany). Incubation overnight at 4 °C with a rabbit polyclonal antibody against p-NF-κB, total NF-κB (p65; Cell Signaling, Danvers, MA, USA), or actin (Sigma-Aldrich, Darmstadt, Germany) were used to determine the protein expression, which was visualized using a polyclonal secondary antibody (Dako, Belgium) and SuperSignal West Femto Substrate (Thermofisher Scientific, Carlsbad, CA, USA). Normalized expression was analyzed by Image Lab Sofware 5.0 (Bio-Rad, Hercules, CA, USA).

### 4.9. Statistical Analyses

Statistical analyses were performed using GraphPad Prism version 6 (GraphPad Software, San Diego, CA, USA). Data were analyzed using the Mann–Whitney U test for unpaired samples and Wilcoxon signed-rank test for paired samples as indicated. Data are expressed as mean ± SEM, and values of * *p* < 0.05, ** *p* < 0.01, and *** *p* < 0.001 were considered statistically significant.

## Figures and Tables

**Figure 1 ijms-24-02311-f001:**
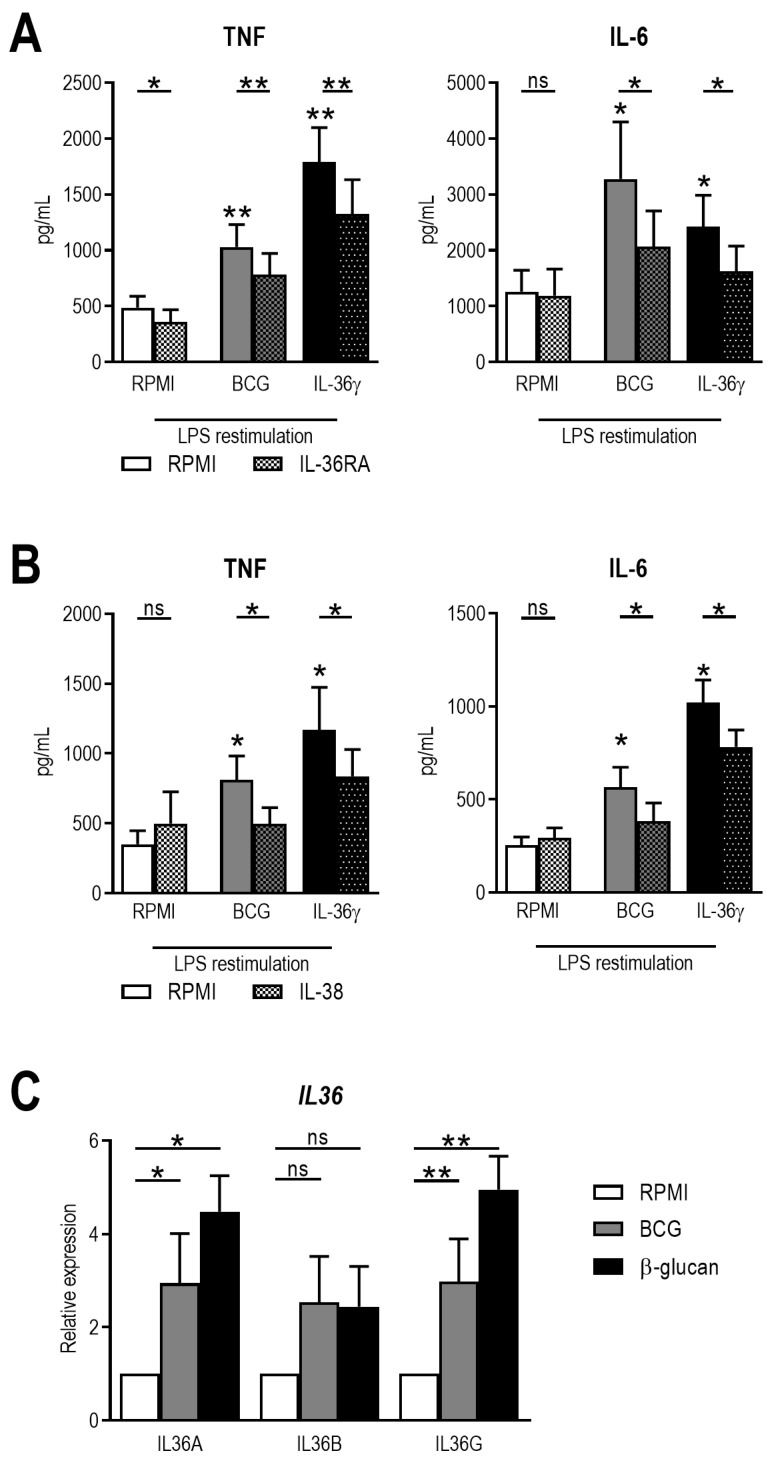
Trained immunity is inhibited by IL-36RA and IL-38. (**A**,**B**) Human monocytes were incubated for 24 h with 2 μg/mL β-glucan or 2 μg/mL of BCG with or without 400 ng/mL IL-36RA (**A**) or 90 ng/mL IL-38 (**B**). After five days of rest in medium, cells were restimulated with 10 ng/mL LPS, and TNF and IL-6 were determined in supernatants. (**C**) Human monocytes were incubated for 4 h with b-glucan or BCG before mRNA isolation and determination of expression isoforms of IL-36. (Mean ± SEM, n = 6, Wilcoxon signed-rank test, ns = non-significant, * *p* < 0.05, ** *p* < 0.01.)

**Figure 2 ijms-24-02311-f002:**
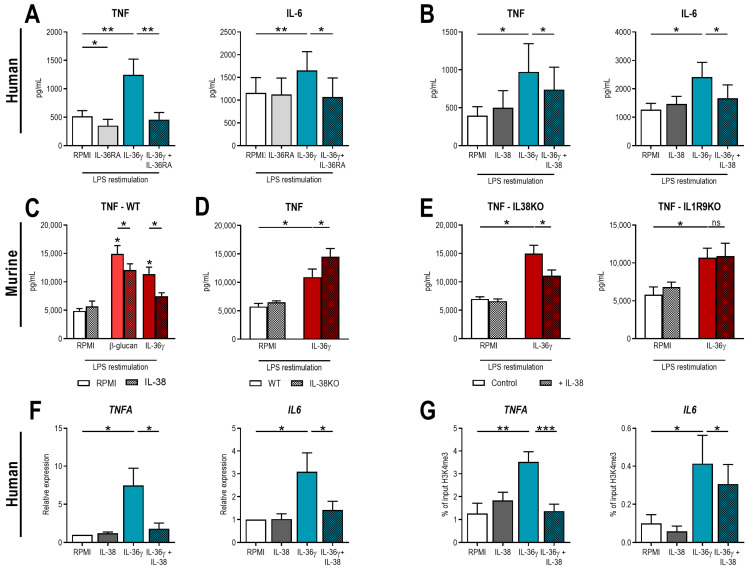
IL-36γ induces trained immunity. (**A**,**B**) Human monocytes were incubated for 24 h with 50 ng/mL of IL-36γ with or without 400 ng/mL IL-36RA (**A**) or 90 ng/mL IL-38 (**B**). After five days of rest in medium, cells were restimulated with 10 ng/mL LPS, and TNF and IL-6 were determined in supernatants. (**C**) Fresh bone marrow of WT mice was stimulated with 2 μg/mL β-glucan or 50 ng/mL IL-36μ in the presence or absence of 90 ng/mL IL-38 for 48 h. After four days of rest, cells were restimulated with 10 ng/mL LPS for 24 h. TNF production was determined in supernatants. (**D**) Fresh bone marrow cells of WT and IL-38KO mice were incubated for 24 h with 50 ng/mL IL-36γ. After four days of rest, cells were restimulated with 10 ng/mL LPS for 24 h. TNF production was determined in supernatants. (**E**) Fresh bone marrow of IL-38KO and IL-1R9KO mice was incubated with 50 ng/mL IL-36γ in the presence or absence of 90 ng/mL IL-38 for 48 h. After four days of rest, cells were restimulated with 10 ng/mL LPS for 24 h. TNF production was determined in supernatants. (**F**) Human monocytes were incubated for 24 h with 50 ng/mL of IL-36γ with or without 90 ng/mL IL-38. After five days of rest in medium, cells were restimulated with 10 ng/mL LPS, and expression of *TNFA* and *IL6* was determined after 4 h. (**G**) Human monocytes were incubated for 24 h with 50 ng/mL of IL-36γ with or without 90 ng/mL IL-38. After five days of rest in medium, cells were fixed, and the percentage of H3K4me3 at the promoters of *TNFA* and *IL6* was determined. (Mean ± SEM, n = 6, Wilcoxon signed-rank test, ns = non-significant, * *p* < 0.05, ** *p* < 0.01, *** *p* < 0.001.)

**Figure 3 ijms-24-02311-f003:**
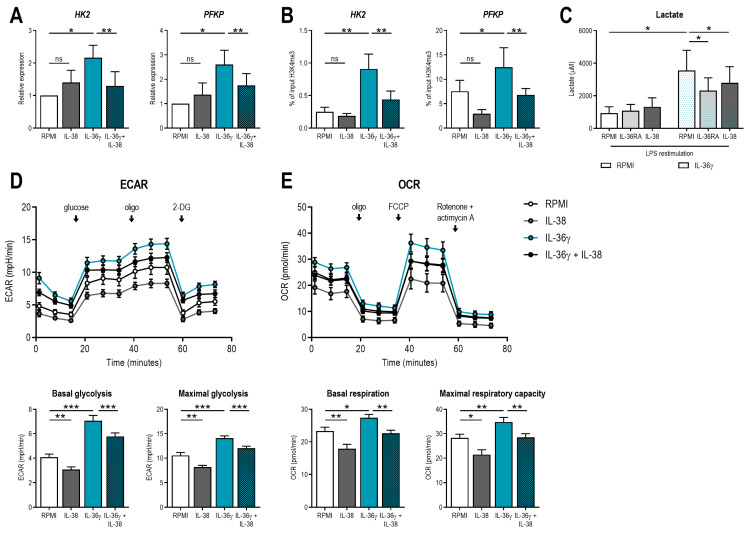
IL-36γ induces immunometabolic changes in human monocytes. (**A**) Human monocytes were incubated for 24 h with 50 ng/mL of IL-36γ with or without 90 ng/mL IL-38. After five days of rest in medium, cells were restimulated with 10 ng/mL LPS, and expression of *HK* and *PFKP* was determined after 4 h. (**B**) Human monocytes were incubated for 24 h with 50 ng/mL of IL-36γ with or without 90 ng/mL IL-38. After five days of rest in medium, cells were fixed, and the percentage of H3K4me3 at the promoters of *HK* and *PFKP* was determined. (**C**) Human monocytes were incubated for 24 h with 50 ng/mL of IL-36γ with or without 90 ng/mL IL-38. After five days of rest in medium, cells were restimulated with 10 ng/mL LPS for 24 h, and lactate concentration was determined in supernatants. (**D**,**E**) Human monocytes were incubated for 24 h with 50 ng/mL of IL-36γ with or without 90 ng/mL IL-38. After five days of rest in medium, extracellular acidification rate (ECAR) (**D**) and oxygen consumption rate (OCR) (**E**) were determined by Seahorse in a glycolysis and mito stress test, respectively. (Mean ± SEM, n = 6, Wilcoxon signed-rank test, ns = non-significant, * *p* < 0.05, ** *p* < 0.01, *** *p* < 0.001.)

**Figure 4 ijms-24-02311-f004:**
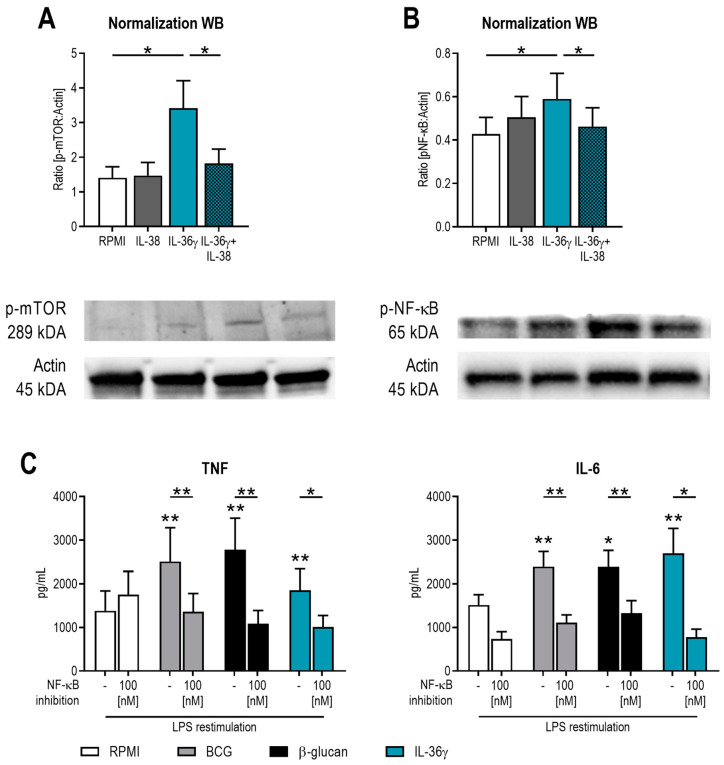
IL-36γ induces phosphorylation of mTOR and NF-κB. (**A**,**B**) Human monocytes were pre-incubated for 1 h with 90 ng/mL IL-38 before 50 ng/mL IL-36γ was added to the medium. Phosphorylation of mTOR was determined after 2 h (**A**), and phosphorylation of NF-κB was determined after 10 min (**B**) by Western blot. (Mean ± SEM, n = 4, Wilcoxon signed-rank test, * *p* < 0.05) (**C**). Human monocytes were incubated for 24 h with 2 μg/mL β-glucan, 2 μg/mL BCG, or 50 ng/mL IL-36γ with or without an inhibitor of NK-κB activation. After five days of rest in 10% human serum, cells were restimulated with LPS, and cytokine production was assessed in supernatants. (Mean ± SEM, n = 6,Wilcoxon signed-rank test, * *p* < 0.05, ** *p* < 0.01, either versus RPMI without NF-κB inhibitor or versus the indicated condition.)

**Table 1 ijms-24-02311-t001:** Primer sequences used for RT-qPCR.

Gene	Forward Primer	Reverse Primer
*HPRT*	CCTGGCGTCGTGATTAGTGAT	AGACGTTCAGTCCTGTCCATAA
*β2M*	ATGAGTATGCCTGCCGTGTG	CCAAATGCGGCATCTTCAAAC
*IL36A*	TTGCCTTAATCTCATGCCGAC	CCGACTTTAGCACACATCAGG
*IL36B*	AGAAATTCAGGGCAAGCCTAC	CAGCCAGGGTAAGAGACTGAC
*IL36G*	GAAACCCTTCCTTTTCTACCGTG	GCTGGTCTCTCTTGGAGGAG
*IL6*	AACCTGAACCTTCCAAAGATGG	TCTGGCTTGTTCCTCACTACT
*TNFA*	AACGGAGCTGAACAATAGGC	TCTCGCCACTGAATAGTAGGG
*β2M*	ATGAGTATGCCTGCCGTGTG	CCAAATGCGGCATCTTCAAAC

**Table 2 ijms-24-02311-t002:** Primer sequences used for ChIP-qPCR.

Gene	Forward Primer	Reverse Primer
*MYO*	AGCATGGTGCCACTGTGCT	GGCTTAATCTCTGCCTCATGAT
*H2B*	TGTACTTGGTGACGGCCTTA	CATTACAACAAGCGCTCGAC
*IL6*	TCGTGCATGACTTCAGCTTT	GCGCTAAGAAGCAGAACCAC
*TNFA*	CAGGCAGGTTCTCTTCCTCT	GCTTTCAGTGCTCATGGTGT
*HK2*	GAGCTCAATTCTGTGTGGAGT	ACTTCTTGAGAACTATGTACCCTT
*PFKP*	CGAAGGCGATGGGGTGAC	ATCTTGCGGGCCACTAGAAG

## Data Availability

The data presented in this study are available on request from the corresponding author.

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
