# Peer review of "Opposing Effects of Interleukin-36γ and Interleukin-38 on Trained Immunity"

_ijms, 2023, doi:10.3390/ijms24032311_

Round 1

Reviewer 1 Report

In this study, we show that IL-36γ induces trained immunity and that IL-38 suppresses signaling from IL-36R, since IL-36 and IL-38 signal through a common receptor (IL-1R6). This is expected to be useful in therapeutic strategies to prevent trained immunity. Experiments used primary human monocytes and showed induction of trained immunity by IL-36RA and suppression by IL-38. The inhibition effect was not as strong but significant. The study has also been confirmed using bone marrow cells from IL-38KO and IL1R9KO mice. The mechanism is shown to correlate with the transcript levels of TNFA and IL6 and the H3K4me3 status of their promoter regions. This reflects with epigenetics associated with trained immunity. In addition, cellular metabolism was also examined as a trained immunity and was consistent with the activation state of HK2 and PFKP. Correlations were also observed with the phosphorylation status of downstream mTOR and NF-κB. These results would provide that inhibition of IL-36 by IL-38 is expected to be effective, at least in trained immune response cells.

As a minor comment, what is the expected level of variation in the human serum used in the experimental system for primary human monocytes?

Author Response

Response to Reviewer 1

Comments and Suggestions for Authors:

In this study, we show that IL-36γ induces trained immunity and that IL-38 suppresses signalling from IL-36R, since IL-36 and IL-38 signal through a common receptor (IL-1R6). This is expected to be useful in therapeutic strategies to prevent trained immunity. Experiments used primary human monocytes and showed induction of trained immunity by IL-36RA and suppression by IL-38. The inhibition effect was not as strong but significant. The study has also been confirmed using bone marrow cells from IL-38KO and IL1R9KO mice. The mechanism is shown to correlate with the transcript levels of TNFA and IL6 and the H3K4me3 status of their promoter regions. This reflects with epigenetics associated with trained immunity. In addition, cellular metabolism was also examined as a trained immunity and was consistent with the activation state of HK2 and PFKP. Correlations were also observed with the phosphorylation status of downstream mTOR and NF-κB. These results would provide that inhibition of IL-36 by IL-38 is expected to be effective, at least in trained immune response cells.

As a minor comment, what is the expected level of variation in the human serum used in the experimental system for primary human monocytes?

Response

We thank the reviewer for the thorough revision of the manuscript. The question raises a valid point. We aimed to circumvent this potential issue by using two pools of human serum derived from 1) 11 individuals and 2) 46 individuals, the sera of whom have been tested for their ability to induce pro-inflammatory cytokines.

The presence of IL-36 in the serum could induce different cytokines such as IL-6 or IL-1 (Yuan ZC, Xu WD, Liu XY, Liu XY, Huang AF, Su LC. Biology of IL-36 Signaling and Its Role in Systemic Inflammatory Diseases. Front Immunol. 2019 Oct 31;10:2532. doi: 10.3389/fimmu.2019.02532. PMID: 31736959; PMCID: PMC6839525.). Therefore, we measured IL-6 (24 hour stimulation) as well as IL-17 (7 day stimulation) in supernatants of buffy coat-derived PBMCs of three donors incubated with the sera of all tested individuals without an additional stimulus. While we cannot say whether these cytokines were induced by IL-36 or by other soluble factors in our testing set-up, serum derived from individuals inducing detectable levels of IL-6 or IL-17 were excluded from the pooled human serum stock used in cell culture.

The presence of IL-38 in the serum was assessed by ELISA in three different pools, none of which had detectable serum concentrations of IL-38.

Reviewer 2 Report

The authors in invitro models evaluated that exposure to IL-36γ results in long-term pro-inflammatory changes in monocytes, which can be inhibited by IL-38.

Although the findings are interesting, the following comments arise:

The abstract needs to describe the main findings. The main findings of the different experiments that were carried out in the study are not described.

Paragraph 18 and 19:  “but also cytokines including IL-1…”. , the statement refers to IL-1 beta or the IL-1 superfamily, review and specify

Paragraph 57 is TNF-alpha?

Figures 1 A and B, 2 A and B, 4C is TNF-alpha?

Figures 1 A and B correct ml by mL, also text in paragraphs 66 to 71 and throughout the document

Material and methods section, mentions how many healthy donors the isolated monocytes were acquired.

Was a dose-response performed to identify the optimal concentration of IL-38 IL-36RA, BCG, beta-glucan, IL-36 gamma, and LPS? Describe in the materials and methods section.

The authors must describe the limitations of the study in the discussion section, supporting the following points: 1) at the methodological level, 2) study design, 3) establishment of causal relationships or associations, as well as 4) the perspectives of the study.

Round 2

Reviewer 2 Report

The authors responded to the requested reviews. The paper can be published in the present form.